# Generative Shape Models: Joint Text Recognition and Segmentation with Very Little Training Data

**Xinghua Lou, Ken Kansky, Wolfgang Lehrach, CC Laan**
Vicarious FPC Inc., San Francisco, USA
`xinghua,ken,wolfgang,cc@vicarious.com`

**Bhaskara Marthi, D. Scott Phoenix, Dileep George**
Vicarious FPC Inc., San Francisco, USA
`bhaskara,scott,dileep@vicarious.com`

## Abstract

Abstract: We demonstrate that a generative model for object shapes can achieve state of the art results on challenging scene text recognition tasks, and with orders of magnitude fewer training images than required for competing discriminative methods. In addition to transcribing text from challenging images, our method performs fine-grained instance segmentation of characters. We show that our model is more robust to both affine transformations and non-affine deformations compared to previous approaches.

## 1 Introduction

Classic optical character recognition (OCR) tools focus on reading text from well-prepared scanned documents. They perform poorly when used for reading text from images of real world scenes [1]. Scene text exhibits very strong variation in font, appearance, and deformation, and image quality can be lowered by many factors, including noise, blur, illumination change and structured background. Fig. 1 shows some representative images from two major scene text datasets: International Conference on Document Analysis and Recognition (ICDAR) 2013 and Street View Text (SVT).

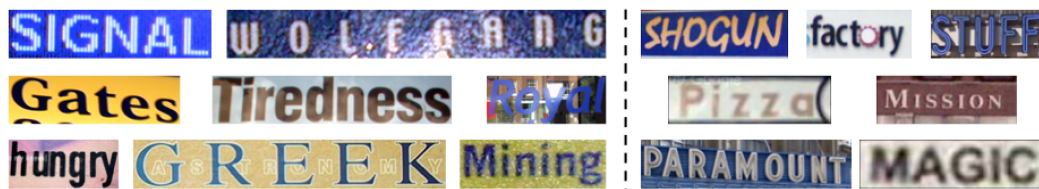

Figure 1: Examples of text in real world scenes: ICDAR 2013 (left two columns) and SVT (right two columns). Unlike classic OCR that handles well-prepared scanned documents, scene text recognition is difficult because of the strong variation in font, background, appearance, and distortion.

Despite these challenges, the machine learning and computer vision community have recently witnessed a surging interest in developing novel approaches for scene text recognition. This is driven by numerous potential applications such as scene understanding for robotic control and augmented reality, street sign reading for autonomous driving, and image feature extraction for large-scale image search. In this paper we present a novel approach for robust scene text recognition. Specifically, we study the problem of text recognition in a cropped image that contains *a single word*, which is usually the output from some text localization method (see [2] for a thorough review on this topic).

Our core contribution is a novel generative shape model that shows strong generalization capabilities. Unlike many previous approaches that are based on discriminative models and trained on millions of real world images, our generative model only requires hundreds of training images, yet still effectively captures affine transformations and non-affine deformations. To cope with the strong variation of fonts in real scenes, we also propose a greedy approach for selecting representative fonts from a large database of fonts. Finally, we introduce a word parsing model that is trained using structured output learning.

We evaluated our approach on ICDAR 2013 and SVT and achieved state-of-the-art performance despite using several orders of magnitude less of training data. Our results show that instead of relying on a massive amount of supervision to train a discriminative model, a generative model trained on uncluttered fonts with properly encoded invariance generalizes well to text in natural images and is more interpretable.

## 2 Related Work

We only consider literature on recognizing scene text in English. There are two paradigms for solving this problem: character detection followed by word parsing, and simultaneous character detection and word parsing.

Character detection followed by word parsing is the more popular paradigm. Essentially, a character detection method first finds candidate characters, and then a parsing model searches for the true sequence of characters by optimizing some objective function. Many previous works following this paradigm differ in the detection and parsing methods.

Character detection methods can be patch-based or bottom-up. Patch-based detection first finds patches of (hopefully) single characters using over-segmentation [3] or the stroke width transformation [4], followed by running a character classifier on each patch. Bottom-up detection first creates an image-level representation using engineered or learned features and then finds instances of characters by aggregating image-level evidence and searching for strong activations at every pixel. Many different representations have been proposed such as Strokelets [5], convolutional neural networks [6], region-based features [7], tree-structured deformable models [8] and simple shape template [9]. Both patch-based and bottom-up character detection have flawed localization because they cannot provide accurate segmentation boundaries of the characters.

Unlike detection, word parsing methods in literature show strong similarity. They are generally sequence models that utilize attributes of individual as well as adjacent candidate characters. They differ in model order and inference techniques. For example, [10] considered the problem as a high-order Markov model in a Bayesian inference framework. A classic pairwise conditional random field was also used by [8, 11, 4], and inference was carried out using message passing [11] and dynamic programming [8, 4]. Acknowledging that a pairwise model cannot encode as useful features as a high-order character n-gram, [3] proposed a patch-based sequence model that encodes up to 4th-order character n-grams and applied beam search to solve it.

A second paradigm is simultaneous character detection and word parsing, reading the text without an explicit step for detecting the characters. For example, [12] proposed a graphical model that jointly models the attributes, location, and class of characters as well as the language consistency of the word they constitute. Inference was carried out using weighted finite-state transducers (WFSTs). [13] took a drastically different approach: they used a lexicon of about 90k words to synthesize about 8 million images of text, which were used to train a CNN that predicts a character at each independent position. The main drawback of this "all-in-one" approach is weak invariance and insufficient robustness, since changes in any attribute such as spacing between characters may cause the system to fail due to over-fitting to the training data.

## 3 Model

Our approach follows the first detection-parsing paradigm. First, candidate characters are detected using a novel generative shape model trained on clean character images. Second, a parsing model is used to infer the true word, and this parser is trained using max-margin structured output learning.

## 3.1 Generative Shape Model for Fonts

Unlike many vision problem such as distinguishing dogs from cats where many *local* discriminative features can be informative, text in real scenes, printed or molded using some fonts, is not as easily distinguishable from *local* features. For example, the curve "‿" at the bottom of "O" also exists in "G", "U" and "Q". This special structure "⊢" can be found in "B", "E", "F", "H", "P" and "R". Without a sense of the global structure, a naive accumulation of local features easily leads to false detections in the presence of noise or when characters are printed tightly. We aim at building a model that specifically accounts for the *global* structure, i.e. the entire shape of characters. Our model is generative such that during testing time we obtain a segmentation together with classification, making the final word parsing much easier due to better explaining-away.

**Model Construction**

During training we build a graph representation from rendered clean images of fonts, as shown in Fig. 2. Since we primarily care about shape, the basic image-level feature representation relies only on edges, making our model invariant to appearance such as color and texture. Specifically, given a clean font image we use 16 oriented filters to detect edges, followed by local suppression that keeps at most one edge orientation active per pixel (Fig. 2a). Then, "landmark" features are generated by selecting one edge at a time, suppressing any other edges within a fixed radius, and repeat (Fig. 2b). We then create a pool variable centered around each landmark point such that it allows translation pooling in a window around the landmark (Fig. 2b). To coordinate the the pool choices between adjacent landmarks (thus the shape of the letter), we add "lateral constraints" between neighboring pairs of pools that lie on the same edge contour (blue dashed lines in Fig. 2c). All lateral constraints are elastic, allowing for some degree of affine and non-affine deformation. This allows our model to generalize to different variations observed in real images such as noise, aspect change, blur, etc. In addition to contour laterals, we add lateral constraints between distant pairs of pixels (red dashed lines in Fig. 2c) to further constrain the shapes this model can represent. These distant laterals are greedily added one at a time, from shortest to longest, between pairs of features that with the current constraints can deform more than $\gamma$ times the deformation allowed by adding a direct constraint between the features (typically $\gamma \approx 3$).

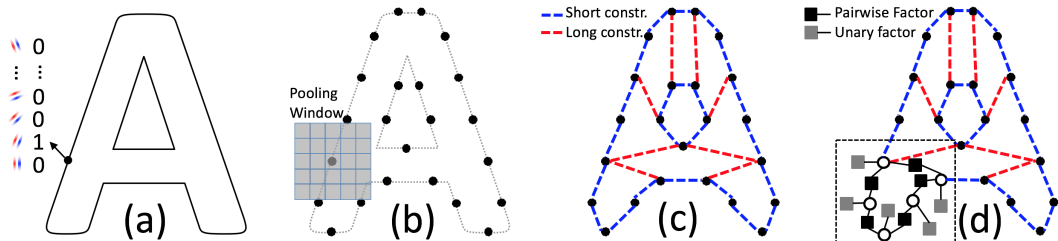

Figure 2: Model construction process for our generative shape model for fonts. Given a clean character images, we detect 16 oriented edge features at every pixel (a). We perform a sparsification process that selects "landmark" features from the dense edge map (b) and then add pooling window (b). We then add lateral constraints to constrain the shape model (c). A factor graph representation of our model is partially shown in (d). (best viewed in color)

Formally, our model can be viewed as a factor graph shown in Fig. 2d. Each pool variable centered on a landmark feature is considered a random variable and is associated with unary factors corresponding to the translations of the landmark feature. Each lateral constraint is a pairwise factor. The unary factors give positive scores when matching features are found in the test image. The pairwise factor is parameterized with a single perturbation radius parameter, which is defined as the largest allowed change in the relative position of features in the adjacent pools. This perturbation radius forbids extreme deformation, giving $-\infty$ log probability if this lateral constraint is violated. During testing, the state space of each random variable is the pooling window and lateral constraints are not allowed to be violated.

During training, this model construction process is carried out independently for all letter images, and each letter is rendered in multiple fonts.

**Inference and Instance Detection** The letter models can be considered to be tiling an input image at all translations. Given a test image, finding all candidate character instances involves two steps: a forward pass and backtracing.

The forward pass is a bottom-up procedure that accumulate evidence from the test image to compute the marginal distribution of the shape model at each pixel location, similar to an activation heatmap. To speed-up the computation, we simplify our graph (Fig. 2c) into a minimum spanning tree, computed with edge weights equal to the pixel distance between features. Moreover, we make the pooling window as large as the entire image to avoid a scanning procedure. The marginals in the tree can be computed exactly and quickly with a single iteration of belief propagation. After non-maximum suppression, a few positions that have the strongest activation are selected for backtracing. This process is guaranteed to overestimate the true marginals in the original loopy graphical model, so this forward pass admits some false positives. Such false positives occur more often when the image has a tight character layout or a prominent texture or background.

Given the estimated positions of character instances, backtracing is performed in the original loopy graph to further reduce false positives and to output a segmentation (by connecting the "landmarks") of each instance in the test image. The backtracing procedure selects a single landmark feature, constrains its position to one of the local maxima in its marginal from the forward pass, and then performs MAP inference in the full loopy graph to estimate the positions of all other landmarks in the model, which provides the segmentation. Because this inference is more accurate than the forward pass, additional false positives can be pruned after backtracing.

In both the forward and backward pass, classic loopy belief propagation was sufficient.

**Greedy Font Model Selection** One challenge in scene text reading is covering the huge variation of fonts in uncontrolled, real world images. It is not feasible to train on all fonts because it is too computationally expensive and redundant. We resort to an automated greedy font selection approach. Briefly, for some letter we render images for all fonts and then use the resulting images to train shape models. These shape models are then tested on every other rendered image, yielding a compatibility score (amount of matching "landmark" features) between every pair of fonts of the same letter. One font is considered representable by another if their compatibility score is greater than a given threshold (=0.8). For each letter, we find and keep the fonts that can represent most other fonts and remove it from the font candidate set together with all the fonts it represents. This selection process is repeated until 90% of all fonts are represented. Usually the remaining 10% fonts are non-typical and rare in real scenes.

### 3.2 Word Parsing using Structured Output Learning

**Parsing Model** Our generative shape models were trained independently on all font images. Therefore, no explaining-away is performed before parsing. The shape model shows high invariance and sensitivity, yielding a rich list of candidate letters that contains many false positives. For example, an image of letter "E" may also trigger the following: "F", "I", "L" and "c". Word parsing refers to inferring the true word from this list of candidate letters.

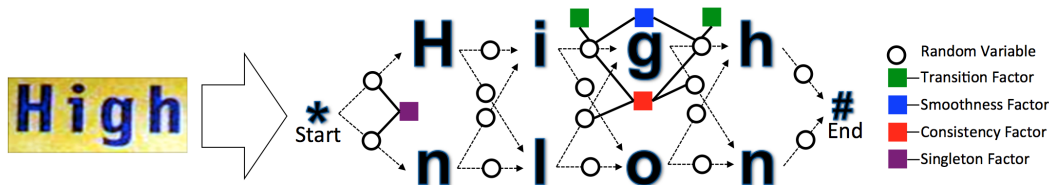

Figure 3: Our parsing model represented as a high-order factor graph. Given a test image, the shape model generates a list of candidate letters. A factor graph is created by adding edges between hypothetical neighboring letters and considering these edges as random variables. Four types of factors are defined: transition, smoothness, consistency, and singleton factors. The first two factors characterize the likelihood of parsing path, while the latter two ensure valid parsing output.

Our parsing model can be represented as a high-order factor graph in Fig. 3. First, we build hypothetical edges between a candidate letter and every candidate letter on its right-hand side within some distance. Two pseudo letters "*" and "#" are created, indicating *start* and *end* of the graph,

respectively. Edges are created from *start* to all possible head letters and similarly from *end* to all possible tail letters. Each edge is considered as a binary random variable which, if activated, indicates a pair of neighboring letters from the true word.

We define four types of factors. Transition factors (green, unary) describe the likelihood of a hypothetical pair of neighboring letters being true. Similarly, smoothness factors (blue, pairwise) describe the likelihood of a triplet of consecutive letters. Two additional factors are added as constraints to ensure valid output. Consistency factors (red, high-order) ensure that if any candidate letter has an activated inward edge, it must have one activated outward edge. This is sometimes referred to as "flow consistency". Lastly, to satisfy the single word constraint, a singleton factor (purple, high-order) is added such that there must be a single activated edge from "start". Examples of these factors are shown in Fig. 3.

Mathematically, assuming that potentials on the factors are provided, inferring the state of random variables in the parsing factor graph is equivalent to solving the following optimization problem.

$$\hat{z} = \ \arg\max_{z} \left\{ \sum_{c \in \boldsymbol{C}} \sum_{v \in \mathrm{Out}(c)} \phi_v^{\mathrm{T}}(\boldsymbol{w}^{\mathrm{T}}) z_v + \sum_{c \in \boldsymbol{C}} \sum_{\substack{u \in \mathrm{In}(c) \\ v \in \mathrm{Out}(c)}} \phi_{u,v}^{\mathrm{S}}(\boldsymbol{w}^{\mathrm{S}}) z_u z_v \right\} \tag{1}$$

$$\text{s.t.} \quad \forall c \in \boldsymbol{C}, \sum_{u \in \mathrm{In}(c)} z_u = \sum_{v \in \mathrm{Out}(c)} z_v, \tag{2}$$

$$\sum_{v \in \mathrm{Out}(*)} z_v = 1, \tag{3}$$

$$\forall c \in \boldsymbol{C}, \forall v \in \mathrm{Out}(c), z_v \in \{0, 1\}. \tag{4}$$

where, $\boldsymbol{z} = \{z_v\}$ is the set of all binary random variables indexed by $v$; $\boldsymbol{C}$ is the set of all candidate letters, and for candidate letter $c$ in $\boldsymbol{C}$, $\mathrm{In}(c)$ and $\mathrm{Out}(c)$ index the random variables that correspond to the inward and outward edges of $c$, respectively; $\phi_v^{\mathrm{T}}(\boldsymbol{w}^{\mathrm{T}})$ is the potential of transition factor at $v$ (parameterized by weight vector $\boldsymbol{w}^{\mathrm{T}}$) and $\phi_{u,v}^{\mathrm{S}}(\boldsymbol{w}^{\mathrm{S}})$ is the potential of smoothness factor from $u$ to $v$ (parameterized by weight vector $\boldsymbol{w}^{\mathrm{S}}$); Constraints (2)–(4) ensure flow consistency, singleton, and the binary nature of all random variables.

**Parameter Learning** Another issue is proper parameterization of the factor potentials, i.e. $\phi_v^{\mathrm{T}}(\boldsymbol{w}^{\mathrm{T}})$ and $\phi_{u,v}^{\mathrm{S}}(\boldsymbol{w}^{\mathrm{S}})$. Due to the complex nature of real world images, high dimensional parsing features are required. For one example, consecutive letters of the true word are usually evenly spaced. For another example, a character n-gram model can be used to resolve ambiguous letter detections and improve parsing quality. We use Wikipedia as the source for building our character n-gram model. Both $\phi_v^{\mathrm{T}}(\boldsymbol{w}^{\mathrm{T}})$ and $\phi_{u,v}^{\mathrm{S}}(\boldsymbol{w}^{\mathrm{S}})$ are linear models of some features and a weight vector. To learn the best weight vector that directly maps the input-output dependency of the parsing factor graph, we used the maximum-margin structured output learning paradigm [14].

Briefly, maximum-margin structured output learning attempts to learn a direct functional dependency between structured input and output by maximizing the margin between the compatibility score of the ground truth solution and that of the second best solution. It is an extension to the classic support vector machine (SVM) paradigm. Usually, the compatibility score is a linear function of some so-called joint feature vector (i.e. parsing features) and feature weights to be learned (i.e. $\boldsymbol{w}^{\mathrm{T}}$ and $\boldsymbol{w}^{\mathrm{S}}$ here). We designed 18 parsing features, including the score of individual candidate letters, color consistency between hypothetical neighboring pairs, alignment of hypothetical consecutive triplets, and character n-grams up to third order.

**Re-ranking** Lastly, top scoring words from the second-order Viterbi algorithm are re-ranked using statistical word frequencies from Wikipedia.

## 4 Experiments

### 4.1 Datasets

**ICDAR** ICDAR ("International Conference on Document Analysis and Recognition") is a biannual competition on text recognition. The ICDAR 2013 Robust Reading Competition was designed for comparing scene text recognition approaches [1]. Unlike digital-born images like those used on the

web, real world image recognition is more challenging due to uncontrolled environmental and imaging conditions that result in strong variation in font, blur, noise, distortion, non-uniform appearance, and background structure. We worked on two datasets: ICDAR 2013 Segmentation dataset and ICDAR 2013 Recognition dataset. In this experiment, we only consider letters, ignoring punctuations and digits. All test images are cropped and each image contains only a single word, see examples in Fig. 1.

**SVT** The Street View Text (SVT) dataset [15] was harvested from Google Street View. Image text in this dataset exhibits high variability and often has low resolution. SVT provides a small lexicon and was created for lexicon-driven word recognition. In our experiments, we did not restrict the setting to a given small lexicon and instead used a general, large English lexicon. SVT does not contain symbols other than letters.

## 4.2 Model Training

**Training Generative Shape Model** To ensure sufficient coverage of fonts, we obtained 492 fonts from Google Fonts[1]. Manual font selection is biased and inaccurate, and it is not feasible to train on all fonts (492 fonts times 52 letters gives 25584 training images). After the proposed greedy font selection process for all letters, we retained 776 unique training images in total (equivalent to a compression rate of 3% if we would have trained on all fonts for all letters). Fig. 4 shows the selected fonts for letter "a" and "A", respectively.

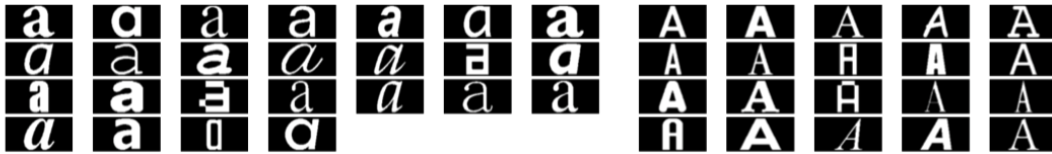

Figure 4: Results of greedy font selection for letter "a" and "A". Given a large font database of 492 fonts, this process leverages the representativeness of our generative shape model to significantly reduce the number of training images required to cover all fonts.

**Training Word Parsing Model** Training the structured output prediction model is expensive in terms of supervision because every training sample consists of many random variables, and the state of every random variable has to be annotated (i.e. the entire parsing path). We prepared training data for our parsing model automatically using the ICDAR 2013 Segmentation dataset [1] that provides per-character segmentation of scene text. Briefly, we first detect characters and construct a parsing graph for each image. We then find the true path in the parsing graph (i.e. a sequence of activated random variables) by matching the detected characters to the ground truth segmentation. In total, we used 630 images for training the parser using PyStruct[2].

**Shape Model Invariance Study** We studied the invariance of our model by testing on transformations of the training images. We considered scaling and rotation. For the former, our model performs robust fitting when the scaling varies between 130% and 70%. For the later, the angle of robust fitting is between -20 and +20 degrees.

## 4.3 Results and Comparison

**Character Detection** We first tested our shape model on the ICDAR 2013 Segmentation dataset. Since this is pre-parsing and no explaining-away is performed, we specifically looked for high recall. A detected letter and a true segmented letter is considered a match only when the letter classes match and their segmentation masks strongly overlap with $\geq 0.8$ IoU (intersection-over-union). Trained on fonts selected from Google Fonts, we obtained a very high 95.0% recall, which is significantly better than 68.7% by the best reported method on the dataset [1]. This attributes to the high invariance encoded in our model from the lateral constraints. The generative nature of the model gives a complete segmentation and classification instead of only letter classification (as most discriminative models do). Fig. 5 shows some instances of letters detected by our model. They exhibit strong variance in font and appearance. Note that two scales ($\times 1$, $\times 2$) are used during testing.

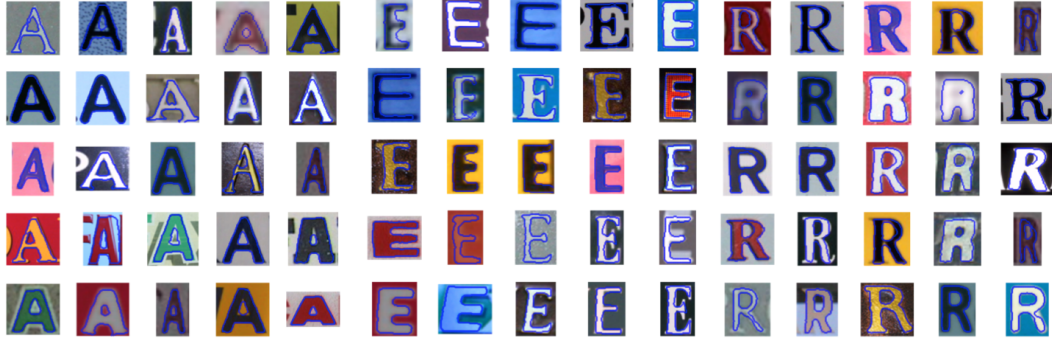

Figure 5: Examples of detected and segmented characters ("A", "E" and "R") from the ICDAR 2013 Segmentation dataset. Despite obvious differences in font, appearance, and imaging condition and quality, our shape model shows high accuracy in localizing and segmenting them from the full image. (best viewed in color and when zoomed in)

**Word Parsing** We compared our approach against top performing ones in the ICDAR 2013 Robust Reading Competition. Results are given in Table 4.3. Our model perform better than Google's PhotoOCR[3] with a margin of 2.3%. However, a more important message is that we achieved this result using three orders of magnitude less training data: 1406 total images (776 letter font images for training the shape models and 630 word images for training the parser) versus 5 million by PhotoOCR. Two major factors attribute to our high efficiency. First, considering character detection, our model demonstrates strong generalization in practice. Data-intensive models like those in PhotoOCR impose weaker structural priors and require significantly more supervision. Second, considering word parsing, our generative model solves recognition and segmentation together, allowing the use of highly accurate parsing features. On the other hand, PhotoOCR's neural-network based character classifier is incapable of generating accurate character segmentation boundaries, making the parsing quality bounded. Our observations on the SVT dataset are similar: using exactly the same training data we achieved state-of-the-art 80.7% accuracy. Note that all reported results in our experiments are case-sensitive. Fig. 6 demonstrates the robustness of our approach toward unusual fonts, noise, blur, and distracting backgrounds.

| Method | ICDAR | SVT | Training Data Size |
|---|---|---|---|
| PicRead [1] | 63.1% | 72.9% | N/A |
| Deep Struc. Learn. [16] | 81.8% | 71.7% | 8,000,000 (synthetic) |
| PhotoOCR [3] | 84.3% | 78.0% | 7,900,000 (manually labeled + augmented) |
| This paper | **86.2%** | **80.7** % | **1,406** (776 letter images + 630 word images) |

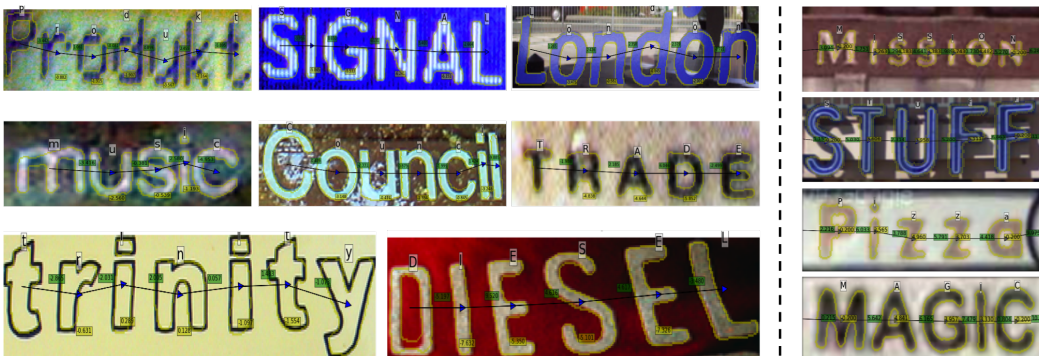

Figure 6: Visualization of correctly parsed images from ICDAR (first two columns) and SVT (last column) including per-character segmentation and parsing path. The numbers therein are local potential values on the parsing factor graph. (best viewed in color and when zoomed in)

### 4.4 Further Analysis & Discussion

**Failure Case Analysis** Fig. 7 shows some typical failure cases for our system (left) and PhotoOCR (right). Our system fails mostly when the image is severely corrupted by noise, blur, over-exposure, or when the text is handwritten. PhotoOCR fails at some clean images where the text is easily readable. This reflects the limited generalization of data-intensive models because of the diminishing return of more training data. This comparison also shows that our approach is more interpretable: we can quickly justify the failure reasons by viewing the letter segmentation boundaries overlaid on the raw image. For example, over-exposure and blur cause edge features to drop out and thus fail the shape model. On the contrary, it is not so straightforward to explain why a discriminative model like PhotoOCR fails at some cases as shown in Fig. 7.

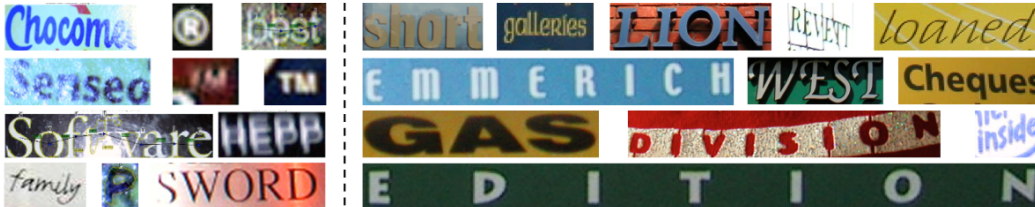

Figure 7: Examples of failure cases for our system and PhotoOCR. Typically our system fails when the image is severely corrupted or contains handwriting. PhotoOCR is susceptible to failing at clean images where the text is easily readable.

**Language Model** In our experiments, a language model plays two roles: in parsing as character n-gram features and in re-ranking as word-level features. Ideally, a perfect perception system should be able to recognize most text in ICDAR without the need for a language model. We turned off the language model in our experiments and observed approximately a $15\%$ performance drop. For PhotoOCR in the same setting, the performance drop is more than $40\%$. This is due to the fact that PhotoOCR's core recognition model is a coarse understanding of the scene, and parsing is difficult without the high quality character segmentation that our generative shape model provides.

**Relation to Other Methods** Here we discuss the connections and differences between our shape model and two very popular vision models: deformable parts models (DPM) [17] and convolutional neural networks (CNN) [18]. The first major distinction is that both DPM and CNN are discriminative while our model is generative. Only our model can generate segmentation boundaries without any additional ad-hoc processing. Second, CNN does not model any global shape structure, depending solely on local discriminative features (usually in a hierarchical fashion) to perform classification. DPM accounts for some degree of global structure, as the relative positions of parts are encoded in a star graph or tree structure. Our model imposes stronger global structure by using short and long lateral constraints. Third, during model inference both CNN and DPM only perform a forward pass, while ours also performs backtracing for accurate segmentation. Finally, regarding invariance and generalization, we directly encode invariance into the model using the perturbation radius in lateral constraints. This is proven very effective in capturing various deformations while still maintaining the stability of the overall shape. Neither DPM nor CNN encode invariance directly and instead rely on substantial data to learn model parameters.

## 5 Conclusion and Outlook

This paper presents a novel generative shape model for scene text recognition. Together with a parser trained using structured output learning, the proposed approach achieved state-of-the-art performance on the ICDAR and SVT datasets, despite using orders of magnitude fewer training images than many pure discriminative models require. This paper demonstrates that it is preferable to directly encode invariance and deformation priors in the form of lateral constraints. Following this principle, even a non-hierarchical model like ours can outperform deep discriminative models.

In the future, we are interested in extending our model to a hierarchical version with reusable features. We are also interested in further improving the parsing model to account for missing edge evidence due to blur and over-exposure.

## Footnotes

[1] https://www.google.com/fonts

[2] https://pystruct.github.io

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
