[Reviews · NeurIPS 2016]

Reviewer 1

Summary

This paper addresses the problem of scene text recognition using a a combination of generative letter shape model and letter sequence parsing model, achieving result competitive with the state of the art but using orders of magnitude fewer examples.

Qualitative Assessment

This is an interesting, well motivated paper with a novel (afaik) model achieving good result. The fact that it learns from a limited amount of data and tries to build a model of the underlying phenomenon (letters + sequence) is attractive from an AI point of view (as opposed to throwing loads of data on a large over-parameterized model, for example). It also helps with interpretability as suggested by Section 4.4. My main concern is that the technical details are sparse. I don't think it would be easy to a motivated reader to reproduce the model in detail as it is mostly described in textual form, apart from the cost for the parsing problem in Eqs. 1-4. Even details on the features used in the sturctured output model are incompletely described. The parsing model is intriguing. The added value is unclear, though, over a baseline ngram LM. While section 4.4 suggests that the ngram model is crucial to good performance, I would like to see how much the *parsing model* brings to the table in addition to a ngram model, which can be trained with very high $n$ on char sequences. Edit: Read other reviews and author reply and updated accordingly. Although it looks like there are several problematic technical issues, I still think this is both an interesting and effective model.

Confidence in this Review

2-Confident (read it all; understood it all reasonably well)


Reviewer 2

Summary

The authors present a generative model for word recognition, which is composed of a character-level model and a word-level model, both of which are implemented as factor graphs. The character-level model is trained on a per-character basis using generated images from several fonts (optimized for variability). Training involves applying a series of edge detectors, identifying pixels of the image as landmarks, and parametrizing the relationships between landmark points using a factor graph. The word-level model captures bigram and trigram dependencies between characters while constraining that a character is chosen for each position. The models are applied to two standard benchmark sets (ICDAR and SVT) and show state of the art results

Qualitative Assessment

Method and Novelty: The authors present a model that has a number of strengths. First, the character-level model is trained on synthetically generated images from a font library, independently of the training corpus. Second, the model converts each training image into a factor graph and learns the spatial relationships between landmarks in each character. This model can readily assign a probability to each candidate character for an image, and the authors provide a description of a two-stage inference algorithm that consists of approximate belief propagation followed by refinement via a backtracking procedure. The candidate characters are then supplied to a word model, which is a fairly standard structured prediction using bigram and trigram features. The generative nature of the model allows a clearer understanding of model operation. This model differs substantially from other models currently in use. Clarity: The paper is incredibly well-written, and guides the reader through all of the major aspects of the method. There are numerous examples and figures that elucidate concepts that would otherwise be difficult to understand. I found that the abstract did not adequately capture the contributions of the paper, and encourage the authors to expand this to help communicate the details of the model and contributions to the reader. Evaluation: The model is evaluated on industry-standard datasets, and achieves superior results. The results are explored in detail, and prominent failure modes are analyzed and used as motivation for future work. Issues: Despite being a strong paper, a number of aspects of the model are still unclear. 1. My understanding is that landmarks are determined independently for each font-character instance - are these landmarks generally consistent between the same character in different fonts? 2. The model uses far less training data, but the training time and test time are never discussed. Given the complexity of the generative model and the number of parameters, it's possible that training time makes this model intractable for use. Furthermore, while discriminative models simply run a predictor for a pattern of pixels, the proposed method must extract landmarks and run inference to find candidate characters. In a world where people expect to point their smartphone at unknown text and get instant results, can this method keep up? 3. This method relies on building generative models from a font library, but may be restricted to Latin scripts. How would such a method be applied to Asian characters where the overlap between structural elements in characters is high?

Confidence in this Review

2-Confident (read it all; understood it all reasonably well)


Reviewer 3

Summary

This paper proposes a generative shape model for scence text recognition, and shows that on two datasets it achieves better performance than the state-of-the art algorithms but requiring order of magnitude fewer training images. While the idea seems to be reasonable, many technical details are omitted or unclear. The shape model is only described at a very high level, leaving me wondering questions like what are the oriented filters, how is local suppression done, how are landmark features produced, and so on. How are the parameters in the potential functions chosen? It is also unclear to me what "backtracing" does and why it works -- Since BP is ran on the full graphical model, isn't this contradicting the earlier claim that forward pass need to be done because inference on the full model is too hard? For the word passing model, it seems to me that it is equivalent to a second-order Markov model (take the letters instead of the edges as variables). This eliminates the higher-order consistency constraints, and makes it possible to do exact Viterbi decoding -- In fact, Viterbi algorithm is mentioned in Re-ranking, so which model is used? I also have concerns on the experiments. I assume the reported accuracies are word recognition accuracies but not letter recognition accuracies, is it? The cited experiment results of PhotoOCR do not match the results in the PhotoOCR paper, why is it so? It is claimed that the model is robust to affine transformations and non-affine deformations, but there is no experiments to validate this.

Qualitative Assessment

I'd encourage the authors to address concerns raised in the reviews.

Confidence in this Review

2-Confident (read it all; understood it all reasonably well)


Reviewer 4

Summary

This aim of this paper is to recognize non-handwritten text in images. The model is generative, and inference proceeds using belief propagation on a factor graph for both finding characters in images and parsing words. Word parsing is based on a combination of bigram and trigram language models. Learning proceeds by learning structural models of characters for fonts, and then by greedily selecting fonts to be modelled. Experiments are performed on two font datasets, where the experimental results are incrementally better than OCR, though with a greatly reduced amount of training examples.

Qualitative Assessment

First, the experimental results seem fine, albeit a incremental. They slightly outperform competing methods. Although it's true the authors achieve good results with less training data than competing methods, their method is also expensive and it doesn't seem easy to make use of more training data anyway. Next, this does not seem to be an actual generative model, but a probabilistic model. Because of how the character recognition is setup (a loopy factor graph), and how parsing is performed, it doesn't seem like one could actually generate text with this model. In other words, sampling would seems prohibitively expensive. It would be more fair to say this is a probabilistic model, but not a generative one for which sampled texts could be drawn. The authors state that the framework can generate segmentation boundaries (line 436); can it really? How do you sample from the factor graphs? There are these big normalizing constants in there, and the model is not a DAG, so sampling doesn't seem easy. How are boundary segmentations generated? It would be helpful to see samples from the model. The inference step seems a bit odd. The authors build a minimum-spanning-tree over the edges of the factor graph. How is this done? Factor graphs don't have weights, so how is this spanning tree being created? I found this unclear. How necessary is the MST? How many edges does the loopy graph have, typically? Are the potentials constructed in such a way that message passing is linear in clique size, or is it more expensive? These details are missing in the paper. Also, the authors state that they have "long constraints" in their factor graph for inferring characters, to help model the structure of characters. These constraints of a parameter (\gamma), and it's unclear how this value was set, and whether its value is important. Is the model/inference sensitive to the choice of parameter value? Another modelling problem is the need to model fonts individually. Why is this necessary? Could there not be a richer model of what characters in general look like? What happens when one tries to perform inference on a font that the system hasn't seen (and hasn't seen anything similar)? Would the system be able to state that it's highly uncertain about the parse it produces for this unseen font? Lastly, there is a factor graph for detecting characters which produces marginals, and another factor graph for word parsing. Why not combine the two into one factor graph to do joint character detection and parsing? Why separate the two stages? Overall, the inference and learning seem ad-hoc, and a lot of details are missing. Inference in a large factor graph is challenging, but from the setup of the paper, there is some machinery in the framework that is complicated and seems unnecessary (eg, dividing the factor graph into two stages). Also, the modelling of characters seems brittle and that in-plane word rotations would cause the system to fail; although rotations could be built into the system in theory, it seems impractical here. The experimental results are incremental. Lastly, the margins are way too small, so there is a lot more content here than would normally be allowed.

Confidence in this Review

2-Confident (read it all; understood it all reasonably well)


Reviewer 5

Summary

This paper presents a method for scene text recognition. The recognition task involves two steps. First, the authors built a generative shape model to characterize the global structure of a single letter under various deformation. Several candidate letters can be extracted by fitting the image to the model. Second, they built a word parsing model to infer the true word from the candidate letters. Specifically, the generative shape model is represented by a factor graph, with landmark features as unary factor, and lateral constraints between neighboring landmarks as pairwise factor. Besides, the parsing model is represented by a high-order factor graph. Experiments show the proposed method outperform state-of-the-art methods while only requires a small amount of data.

Qualitative Assessment

Strength: 1. This paper proposes a novel generative shape model to capture the global structure information of different characters. The structure information is invariant to deformations, fonts, appearance, etc. Therefore, it has much better generalization capability and requires orders of magnitude less training data than competing approaches. 2. A greedy approach is proposed to select representative fonts, which further reduce the amount of training data. 3. A high-order factor graph model trained by structured output learning is used for word parsing. Experimental results demonstrate its preferable effectiveness. Weakness: 1. The proposed method consists of two major components: generative shape model and the word parsing model. It is unclear which component contributes to the performance gain. Since the proposed approach follows detection-parsing paradigm, it is better to evaluate on baseline detection or parsing techniques sperately to better support the claim. 2. Lacks in detail about the techniques and make it hard to reproduce the result. For example, it is unclear about the sparsification process since it is important to extract the landmark features for following steps. And how to generate the landmark on the edge? How to decide the number of landmark used? What kind of images features? What is the fixed radius with different scales? How to achieve shape invariance, etc. 3. The authors claim to achieve state-of-the-art results on challenging scene text recognition tasks, even outperforms the deep-learning based approaches, which is not convincing. As claimed, the performance majorly come from the first step which makes it reasonable to conduct comparisons experiments with existing detection methods. 4. It is time-consuming since the shape model is trained in pixel level(though sparsity by landmark) and the model is trained independently on all font images and characters. In addition, parsing model is a high-order factor graph with four types of factors. The processing efficiency of training and testing should be described and compared with existing work. 5. For the shape model invariance study, evaluation on transformations of training images cannot fully prove the point. Are there any quantitative results on testing images?

Confidence in this Review

2-Confident (read it all; understood it all reasonably well)


Reviewer 6

Summary

The authors present a generative shape model for robust text recognition in real world scenes. The paper presents the model, an approximate inference and detection method at character-level and a parsing method at word-level, parameter learning with Structured SVMs, together with experiments and evaluation on real-world datasets, and a discussion of the strength of this generative approach.

Qualitative Assessment

The results look promising, especially given the limited number of training samples. The patches proposed for the "dangerous" aspects (model complexity vs. inference tractability, number of training fonts) are simple yet efficient. It is interesting to see this generative Loopy model method outperforming CNN-based approaches for this task. (1) If I understand well, the messages sent during belief propagation are of tractable size because we restrict the number of candidate pixels for a node, given the type of potential used (-inf outside domain). Maybe the incidence of the choice of this potential for the tractability of the inference should be made clear. Moreover, in DPMs, Generalized Distance Transforms or Branch-and-Bound algorithms are typically used to model more complex pairwise potentials (quadratic). Is this a possible extension of this model? (2) For better understanding of the potential impact, I would appreciate an analysis of the computational requirements for the inference method (time scales of the different steps of the pipeline). (3) I am also curious of the performance of the non-loopy model alone: is the loopy backtracking step necessary for the recognition of characters? How much time does this step take, and is it worth it ? (4) Do you run the model at a single standard scale? Would running the model at different scales help ? I would like these questions answered, especially question (3), to better understand the relevance of the approach. Scores explanation: - Technical quality: model and experiments look sound - Novelty/originality: interesting non-deep contribution - Potential impact or usefulness: could be fine but I would like to know more about questions (2) and (3) - Clarity and presentation are alright. == post-rebuttal update == I find the results and methods of the paper promising. However the rebuttal did not address all concerns raised by the reviewers. - Experiments should support the claim that every step of the pipeline is necessary - Details on the potentials used and the tractability of message passing are missing, as well as a discussion of these potentials (related to DPMs for instance) Omitted details in the paper hinders the technical quality of the paper and its reproducibility. I would invite the authors to add more experiments justifying their model choices and the performance gain they provide, and add more details, to ideally make a reimplementation possible from the paper alone (e.g. mentioning that a multi-scale model is used).

Confidence in this Review

2-Confident (read it all; understood it all reasonably well)